# Molecular Viral Diagnosis and Sanitation of Yam Genetic Resources: Implications for Safe Yam Germplasm Exchange

**DOI:** 10.3390/v12101101

**Published:** 2020-09-29

**Authors:** Marie Umber, Denis Filloux, Suzia Gélabale, Rose-Marie Gomez, Armelle Marais, Séverine Gallet, Franciane Gamiette, Claudie Pavis, Pierre-Yves Teycheney

**Affiliations:** 1Institut National de Recherche pour l’Agriculture, l’Alimentation et l’Environnement, Unité de Recherche Agrosystèmes Tropicaux, F-97170 Petit-Bourg, France; suzia.gelabale@inrae.fr (S.G.); rose-marie.gomez@inrae.fr (R.-M.G.); severine.gallet1@gmail.com (S.G.); franciane.gamiette@gageko.com (F.G.); claudie.pavis@gmail.com (C.P.); 2Centre de Coopération Internationale en Recherche Agronomique Pour le Développement, Unité Mixte de Recherche—Biologie et Génétique des Interactions Plante-Parasite, F-34398 Montpellier, France; denis.filloux@cirad.fr; 3Biologie et Génétique des Interactions Plante-Parasite, Univ. Montpellier, Centre de Coopération Internationale en Recherche Agronomique Pour le Développement, Institut National de Recherche pour l’Agriculture, Montpellier SupAgro, F-34060 Montpellier, France; 4Institut National de Recherche pour l’Agriculture, l’Alimentation et l’Environnement, Université de Bordeaux, Unité Mixte de Recherche Biologie du Fruit et Pathologie, F-33882 Villenave d’Ornon, France; armelle.marais-colombel@inrae.fr; 5Centre de Coopération Internationale en Recherche Agronomique Pour le Développement, Unité Mixte de Recherche Amélioration Génétique et Adaptation des Plantes Méditerranéennes et Tropicales, F-97130 Capesterre Belle-Eau, France; pierre-yves.teycheney@cirad.fr; 6Amélioration Génétique et Adaptation des Plantes Méditerranéennes et Tropicales, Univ. Montpellier, Centre de Coopération Internationale en Recherche Agronomique Pour le Développement, Institut National de Recherche pour l’Agriculture, l’Alimentation et l’Environnement, Institut Agro, F-97130 Capesterre Belle-Eau, France

**Keywords:** molecular diagnosis, yam, viruses, sanitation, germplasm conservation and dissemination

## Abstract

Yam (*Dioscorea* spp.) is an important crop in tropical and subtropical regions. Many viruses have been recently identified in yam, hampering genetic conservation and safe international exchanges of yam germplasm. We report on the implementation of reliable and cost-effective PCR-based detection tools targeting eight different yam-infecting viruses. Viral indexing of the *in vitro* yam collection maintained by the Biological Resources Center for Tropical Plants (BRC-TP) in Guadeloupe (French West Indies) unveiled a high prevalence of potyviruses, badnaviruses, Dioscorea mosaic associated virus (DMaV) and yam asymptomatic virus 1 (YaV1) and a high level of coinfections. Infected yam accessions were subjected to a combination of thermotherapy and meristem culture. Sanitation levels were monitored using PCR-based and high-throughput sequencing-based diagnosis, confirming the efficacy and reliability of PCR-based detection tools. Sanitation rates were highly variable depending on viruses. Sixteen accessions were successfully sanitized, paving the way to safe yam germplasm exchanges and the implementation of clean seed production programs worldwide.

## 1. Introduction

Yam is a staple food in the tropics and subtropics and the second most important tuber crop in Africa after cassava. Annual worldwide yam production was 73 million tons in 2018, with five African countries accounting for 92% of this production and Nigeria being the largest producer with 48 million tons [1]. Although worldwide yam production is increasing, yields are declining in some areas mostly due to decreasing soil fertility or attacks by pests and pathogens. In Guadeloupe (French West Indies), yam production has decreased by almost two thirds in 20 years, from 9000 t in 1997 to 3425 t in 2018 [1]. Hence, Guadeloupe is no longer self-sufficient and must import 80% of its yam tuber consumption needs. This steep decrease results from a combination of socio-economic and health factors such as the decrease of available cultivable lands, the increase in labor costs and the spread of pests and pathogens [2], especially viruses. For example, the American yam species *Dioscorea trifida*, originating from the Amazonian basin and once widely grown for local consumption, is almost no longer cultivated in Guadeloupe due to its high sensitivity to viruses [3]. Likewise, viral diseases strongly affect the production of African yam, *D. rotundata*, in West Africa, where it is a very popular crop [4] and plays an essential role in food security.

The overall high impact of viral diseases on yam production prompted efforts to identify yam-infecting viruses in order to design and implement control strategies based on accurate diagnosis and the production of clean seeds. These efforts led to the characterization of several novel viruses and/or the sequencing of the complete genomes belonging to genera *Ampelovirus*, *Aureusvirus*, *Badnavirus*, *Carlavirus*, *Comovirus*, *Cucumovirus*, *Fabavirus*, *Macluravirus*, *Potexvirus* and *Potyvirus*. Thus, within the last five years alone, the complete genomes of 13 novel yam viral species were sequenced: one potyvirus, *Dioscorea mosaic virus* (DMV, [5]); six badnaviruses, *Dioscorea bacilliform AL virus 2* (DBALV2, [6]), *Dioscorea bacilliform ES virus* (DBESV, [6]), *Dioscorea bacilliform RT virus 1* (DBRTV1, [7]), *Dioscorea bacilliform RT virus 2* (DBRTV2, [7]), Dioscorea bacilliform RT virus 3 (DBRTV3, [8]) and *Dioscorea bacilliform TR virus* (DBTRV, [9]); one sadwavirus, *Dioscorea mosaic associated virus* (DMaV, [10]); two macluraviruses, *Yam chlorotic mosaic virus* (YCMV, [11]) and *Yam chlorotic necrosis virus* (YCNV, [12]); two ampeloviruses, *Air potato ampelovirus 1* (AiPoV1, [5]) and yam asymptomatic virus 1 (YaV1, [13]) and one virus belonging to the family *Betaflexiviridae*, yam virus Y (YVY, [14]).

With more and more yam-infecting viruses being discovered, determining the viral status of yam germplasm has become increasingly challenging and critical to securing the safe production and exchange of genetic resources and preventing the spread of viral diseases. Indeed, international regulations prohibit the circulation of infected biological resources to avoid pathogenic outbreaks. For this, accurate and sensitive diagnosis tools and methods need to be developed and applied. Such tools and methods are also instrumental for the production of clean (virus-free) planting material produced by sanitation programs such as the one implemented by Guadeloupe’s Biological Resource Center for Tropical Plants (BRC-TP). This program aims at producing virus-free yam plants through sanitation with respect to the viruses identified in Guadeloupe (Table 1). This germplasm collection includes accessions belonging to the five most cultivated tropical yam species (*D. alata*, *D. bulbifera*, *D. cayenensis-rotundata*, *D. esculenta* and *D. trifida*).

Several techniques have proved efficient in eliminating viruses from yams, such as chemotherapy, thermotherapy and cryotherapy associated with cell culture [21,22]. Alternative methods have been tested, like hot water treatment on vine cuttings, which resulted in 90% of YMMV elimination [23], or more recently, treatment with ozone dissolved in water, which resulted in 63.5% of potyvirus elimination [24]. The BRC-TP has been using a combination of thermotherapy and meristem culture to eliminate YMV for almost 30 years [3]. However, virus elimination has been monitored by ELISA, a much less sensitive detection technique than commonly used RT-PCR-based detection. Hence, the success rate of this sanitation program needs to be assessed.

Here we report on the development of reliable and cost-effective molecular diagnostic tools for indexing all viruses occurring in yams in Guadeloupe and their implementation for virus prevalence studies. We present the first overall assessment of the efficiency of a yam sanitation process combining thermotherapy and meristem excision and the successful production of sanitized virus-free yam plants.

## 2. Materials and Methods

### 2.1. Plant Material

Plant material originated from the BRC-TP yam collection conserved *in vitro* in Guadeloupe and maintained in sterile conditions on solid growth medium (Table 2) at 24 °C under a photoperiod of 12 h [25]. A total of 396 accessions were used, including 140 accessions of *D. alata*, 4 *D. bulbifera*, 68 *D. cayenensis-rotundata*, 8 *D. esculenta*, 1 *D. hurtiflora*, 1 *D. minutiflora*, 1 *D. nummularia*, 1 *D. pilimpika* and 172 *D. trifida*. A detailed list of this plant material is provided in Appendix A.

### 2.2. Design of Primers for the Detection of DMaV, Yam Macluraviruses and Yam Potexviruses

Nucleotide alignments were built from publicly available DMaV and yam macluraviruses genomic and EST sequences using the CLUSTALW component of Geneious (Biomatters Ltd., Auckland, New Zealand) in order to identify conserved regions in the genome of these viruses (Appendix A). Primers were designed in conserved viral sequences encoding the RNA-dependent RNA polymerase (RdRp) of DMaV and the coat protein of the three yam macluraviruses (Table 3). Primers’ specificity was monitored by direct sequencing of amplification products (Genewiz, Leipzig, Germany).

Likewise, the nucleotide sequences of 16 RNA-dependent RNA polymerase (RdRp) domains from isolates of the three potexvirus species reported in yams [15] were aligned as described above (Appendix A). Five primers were designed in the conserved RdRp domain: three forward primers, YamX-1F, -2F and -3F; and two reverse primers, YamX-4R and -5R (Table 3). The specificity of primers YamX-3F and YamX-5R (Eurogentec, Angers, France) was assessed by sequencing amplification products generated from five infected plants using these primers (Appendix A), following their cloning into pGEM-T easy vector (Promega, Charbonnières, France). A multiple alignment was performed on the sequences of the nine recombinant clones generated (deposited in GenBank under accession numbers MT632008-MT632016) and the corresponding 208 nt sequences described by Acina-Mambole et al. [15] using the CLUSTALW component of MEGA X [34]. Phylogenetic trees were constructed using the Maximum Likelihood method.

### 2.3. Extraction of Total Nucleic Acids and Synthesis of cDNAs 

Total nucleic acids (TNAs) were extracted from 100 mg leaf samples collected from yam plants according to the procedure 2 described by Foissac et al. [35], using a bead homogenizer (MP Biomedicals, Illkirch, France). cDNAs were generated as follows. First, 5 µL of TNAs was denaturated for 5 min at 70 °C, followed by a 5 min incubation on ice with 250 nM of oligonucleotide PolyN_6_ and 125 nM of oligonucleotide PolyT_18_. Then, cDNA was synthetized for 1 h at 37 °C with 500 µM of dNTPs, 20 U of RNAse inhibitor and 200 U of the M-MLV reverse transcriptase (Promega).

### 2.4. Detection of Yam Macluraviruses, CMV, DMaV, YaV1, YMV and YMMV by PCR

PCR reactions were performed on 3 µL of cDNAs with 200 µM of dNTPs and 1.25 U GoTaq G2 polymerase (Promega). Primers and MgCl_2_ concentrations as well as annealing temperatures varied depending on the targeted virus and are detailed in Table 3. PCR conditions were an initial denaturation step of 2 min at 95 °C and 30 cycles of 30 s at 95 °C, 30 s at annealing temperature and 30 s at 72 °C, followed by an elongation step of 5 min at 72 °C, with appropriate control samples (infected and non-infected plants). Amplification products were analyzed by electrophoresis on a 1.2% agarose gel. 

### 2.5. Detection of Yam Potexviruses by Nested PCR

A first PCR was performed on 3 µL of cDNAs using the generic potexvirus-specific primer pair Potex-2RC/Potex-5 described by van der Vlugt and Berendsen [31]. A nested PCR was then performed on 1 µL of the amplification products using YamX-3F/YamX-5R primers. PCR conditions were the same as those described above, and primers and MgCl_2_ concentrations as well as annealing temperatures are provided in Table 3. Amplification products were analyzed by electrophoresis on a 1.5% agarose gel.

### 2.6. Detection of Badnaviruses by Multiplex-Immunocapture-PCR (M-IC-PCR) 

Badnavirus indexing was performed according to Le Provost et al. [36] with the following modifications. Microtubes (0.2 µL) were coated with diluted polyclonal antibodies [37] during 48 h at 4 °C, immunocapture was performed for 4 h at room temperature and samples were treated with 2 U of RQ1 DNAse (Promega) prior to PCR as described by Gambley et al. [38]. Multiplex-PCR was performed using generic badnavirus-specific primer pair BadnaFP/BadnaRP [32] and plant-genomic primer pair atpB1/atpB2 [33], with appropriate control samples (infected, non-infected plants and yam DNA), as described by Umber et al. [9] (Table 3). Amplification products were analyzed by electrophoresis on a 1% agarose gel.

### 2.7. Sanitation Process

Infected accessions were selected from the BRC-TP *in vitro* yam collection. They harbored various combinations of coinfection by yam viruses (DMaV, YaV1, YMV, YMMV, badnaviruses, macluraviruses and potexviruses; Appendix A). Each infected accession was multiplied by cuttings into 24 vitroplants: 20 vitroplants were submitted to a first cycle of sanitation by thermotherapy and meristem culture (see below), and four were conserved as control mother plants. Regenerated vitroplants, called meriplants, were indexed for DMaV, YaV1, YMV, YMMV, badnaviruses, macluraviruses and potexviruses in order to assess the efficacy of the sanitation process. Then, virus-free meriplants were multiplied by cuttings, and two clones of each were acclimatized (see below) and indexed again for DMaV, YaV1, YMV, YMMV, badnaviruses, macluraviruses and potexviruses to ensure complete viral sanitation. Partially sanitized meriplants, i.e., still infected by one or more viruses, were submitted to additional sanitation cycles until the complete elimination of viruses. The sanitation process is detailed in Figure 1.

### 2.8. Thermotherapy and Meristem Excision of Yam Vitroplants

Twenty shoots of selected infected vitroplant accessions were first maintained for two months on solid growth medium (Table 2 and Figure 2A) under sterile conditions at 34 °C and a photoperiod of 12 h in a thermal chamber (ST1 classic, Pol-EKO, Wodzisław Śląski, Poland). Then, 1 mm-sized apical meristems were excised under stereo microscope (Stemi 305, Zeiss, Marly-le-Roi, France) and grown in small Petri dishes on regeneration medium (Table 2) at 26 °C under a 12 h photoperiod (Figure 2B). Plantlets were transferred to growth medium in individual culture tubes upon the emergence of the first leaves and roots (Figure 2C). Once meriplants had grown at least four internodes, 100 mg leaf samples were collected twice in two tubes containing lysis matrix A (MP Biomedicals, Illkirch, France) and stored at −80 °C until viral indexing.

### 2.9. Acclimatization of Yam Vitroplants

Yam vitroplants were removed from tubes, cleaned from culture medium in water with a few drops of detergent and planted in small pots (7 × 7 cm) filled with wet sterile potting soil 110/60/140 N/P/K (Plantaflor, Vechta, Germany). Pots were placed under homemade Plexiglas boxes in order to maintain moisture at as high a level as possible in a room with controlled conditions (12 h photoperiod, 26 °C). After 10 days, boxes were opened slightly, and after another 10 days, completely removed. When yam plantlets had grown approximately 20 leaves (about two months), two young well-developed leaves were collected, placed separately in two grinding bags (Universal 12 × 15 cm, Bioreba, Reinach, Switzerland) and stored at −80 °C until viral indexing.

### 2.10. Assessment of Viral Sanitation by High-Throughput Sequencing-Based Viral Indexing

Following acclimatization, two leaves from each of four sanitized *D. trifida* plantlets (278/TM1, 281/TM1, 528/TM1 and 650/TM2) were collected and processed independently. Double-stranded RNAs (dsRNAs) were purified from leaf samples and used for random amplification as described previously by Marais et al. [39]. Amplification products were then sequenced on the GeT-PlaGe platform (INRAE, Toulouse, France) for MiSeq sequencing (2 × 250 nt) in a multiplex scheme. 

Sequencing reads were cleaned, demultiplexed [40] and assembled into contigs using the VirAnnot pipeline [41] and CLC Genomics Workbench 8.0 (Qiagen, Courtabœuf, France). Annotations were performed based on BLASTn and BLASTx comparisons [42]. Reads were also mapped on a selection of publicly available yam virus sequences retrieved from GenBank. 

## 3. Results

### 3.1. Implementation and Optimization of Viral Detection Tools

In order to assess the viral status of the BRC-TP *in vitro* collection, we developed molecular diagnosis tools targeting yam-infecting viruses present in Guadeloupe (Table 1). Viruses with RNA genomes were detected by RT-PCR performed on total nucleic acids. For the detection of CMV, YaV1, YMV and YMMV, existing primers were used (Table 3). In contrast, no primers were readily available for the detection of DMaV and yam macluraviruses. Therefore, multiple alignments were built using sequences publicly available from GenBank, primers were designed in conserved domains (Appendix A and Table 3) and used successfully for the detection of these viruses.

The use of existing generic primer pair Potex-2RC/Potex-5 [31] for the detection of yam potexviruses by RT-PCR resulted in nonspecific amplification products. Therefore, three forward primers (YamX-1F, YamX-2F, YamX-3F) and two reverse primers (YamX-4R, YamX-5R) were designed in the 584-nt region corresponding to the RdRp domain targeted by the primers designed by van der Vlugt and Berendsen [31], based on multiple alignments of *yam virus X* (YVX), yam potexvirus 1 and yam potexvirus 2 sequences (Appendix A and Table 3). All primer combinations were tested. The most sensitive and reproducible results were obtained using primer pair YamX-3F and YamX-5R in nested PCR performed on an aliquot of RT-PCR products obtained using primer pair Potex-2RC/Potex-5. A selection of amplification products generated by YamX-3F/YamX-5R primers from two *D. cayenensis-rotundata* and five *D. trifida* accessions were cloned and sequenced (Appendix A). The nine obtained sequences were compared to yam potexvirus sequences publicly available from GenBank, showing that they were most closely related to those of *Yam virus X* species (Appendix A).

The detection of badnaviruses was performed by M-IC-PCR to avoid false positives resulting from the amplification of endogenous Dioscorea bacilliform viruses (eDBVs), which are widespread in yam genomes [43].

All viral detection methods used in this work were optimized in order to increase their sensitivity and reliability, leading to optimal MgCl_2_ and primer concentrations as well as annealing temperatures, which are summarized in Table 3.

### 3.2. Assessment of the Sanitary Status of the *In Vitro* BRC-TP Yam Collection

Viral indexing was performed on 396 yam accessions of the *in vitro* yam collection conserved by the BRC-TP, using tools and methods described above for the detection of CMV, DMaV, YaV1, YMV, YMMV, badnaviruses, macluraviruses and potexviruses. The majority of sampled plants belonged to the five most cultivated yam species in Guadeloupe, *Dioscorea alata* (140 sampled plants), *D. bulbifera* (4), *D. cayenensis-rotundata* (68), *D. esculenta* (8) and *D. trifida* (172). Complete indexing results are provided in Appendix A, and Figure 3A shows the prevalence of yam viruses registered for the 396 tested accessions and for the three most represented yam species in the collection, *D. alata*, *D. cayenensis-rotundata* and *D. trifida*.

All targeted viruses were present in the *in vitro* BRC-TP collection, except CMV, despite samples being screened for the presence of this virus in separate experiments using two sets of primers that can detect strains and isolates of both CMV subgroups 1 and 2 [44,45]. Overall, YMMV was the most prevalent virus in the collection, with 57.8% (229/396) infected accessions. In contrast, macluraviruses displayed the lowest prevalence in the same collection, with only 7.1% (28/396) infected accessions. Prevalence for DMaV, YaV1, YMV and badnaviruses in the collection were very similar at 35.4% (140/396), 37.4% (148/396), 36.4% (144/396) and 39.2% (154/393), respectively. 

Mixed infections by up to five distinct viruses were reported in 282 accessions (71.2%; Figure 3B, Appendix A), whereas 33 accessions (8.3%) were free from all targeted viruses. A majority of the indexed accessions was infected by two viruses (29.5%, 117/396), whereas only a few (11/396; 2.8%) were infected by five different viruses. Most of the coinfections involved combinations of YMV and YMMV, YMMV and badnaviruses, YMMV and YaV1, YMMV and DMaV or DMaV and badnaviruses.

### 3.3. Sanitation of the BRC-TP *In Vitro* Yam Collection

Viral detection tools were used to monitor the efficiency of sanitation. A total of 57 infected mother vitroplants from 10 accessions of *D. alata*, four of *D. cayenensis-rotundata* and 43 of *D. trifida,* representing all combinations of co-infection, were subjected to a sanitation cycle: 18 accessions were infected by DMaV, 10 by YaV1, 20 by YMV, 49 by YMMV, 26 by badnaviruses, six by macluraviruses and 12 by potexviruses (Appendix A). Most accessions (41/57) were infected by more than one virus. Following thermotherapy and meristem culture, 285 meriplants were obtained, i.e., an average of five meriplants generated per mother plant, with a minimum of one and a maximum of 13 meriplants per mother plant (Appendix A). On average, *D. cayenensis-rotundata* accessions showed the highest regeneration level (9.8), *D. trifida* the lowest (4.1) and *D. alata* an intermediate level (7.1). 

Each meriplant and its mother plant were indexed twice for DMaV, YaV1, YMV, YMMV, badnaviruses, macluraviruses and potexviruses. No CMV indexing was performed since this virus could not be detected in the BRC-TP *in vitro* collection nor in the fields in Guadeloupe during viral prevalence studies [46]. After one cycle of sanitation, 44.9% of the meriplants (128/285, from 31 accessions) were partially sanitized and 14.4% (41/285, from 17 accessions) were completely free from targeted viruses (Appendix A). 

Twelve non-sanitized or partially sanitized meriplants were subjected to a second cycle of sanitation (Appendix A). A total of 66 meriplants were generated, leading to an average regeneration rate of 5.5 meriplants per accession. This second cycle of sanitation was less successful than the first one, leading to 48.5% partially sanitized meriplants (32/66, from five accessions) and only 4.5% virus-free meriplants (3/66, from two accessions), leading to a total of 44 virus-free meriplants.

### 3.4. Assessment of the Sanitation in Acclimatized Plants 

Viral titers can be low in vitroplants, possibly below detection thresholds [47]. Therefore, sanitation of yam meriplants was further assessed in fully developed plants following acclimatization of vitroplants. For this, two clones of each fully sanitized meriplant were acclimatized and grown under controlled conditions, then indexed for DMaV, YaV1, YMV, YMMV, badnaviruses, macluraviruses and potexviruses once they reached a 20 leaves stage. Five meriplants died in the process, therefore 39 of the 44 sanitized meriplants could be assessed. Indexing results were identical for each meriplant pair and negative for all viruses for 36/39 meriplants corresponding to four *D. alata* accessions, one *D. cayenensis-rotundata* accession and 11 *D. trifida* accessions (Table 4). The three remaining meriplants were still infected by badnaviruses, DMaV or YMV, respectively (Appendix A).

In order to confirm the results obtained using RT-PCR-based diagnostic, the virome of four fully sanitized *D. trifida* accessions was monitored by high throughput sequencing (HTS) of dsRNA [48]. For this, leaf samples were collected from the two acclimatized clones of these four accessions at the 20 leaves stage and analyzed. Prior to sanitation, these accessions were infected by YMV (accession 528), YMMV (accession 281), DMaV and YMMV (accession 278) or YMV, YMMV and badnaviruses (accession 650), respectively (Appendix A). The number of demultiplexed cleaned reads was 19,378 for meriplant 278/TM1, 122,160 for meriplant 281/TM1, 102,474 for meriplant 528/TM2 and 92,674 for meriplant 650/TM1, respectively. After reads assembly and BLAST-based annotation of generated contigs, no contig related to ssRNA+ viruses that were initially present in the four yam accessions (YMV, YMMV and DMaV) could be identified. The analyses made by mapping of reads on YMV, YMMV and DMaV sequences were consistent with this result. Nevertheless, depending on samples, between 0 and 15 contigs were found to be related to mycoviruses or bacteriophages. The lack of sequences related to ssRNA+ viruses leads to the conclusion that the analyzed accessions were successfully sanitized for RNA viruses and confirms molecular indexing results.

Sanitation success rate is defined as the ratio of sanitized meriplants divided by the number of meriplants generated from an infected vitroplant. Sanitation success rates obtained in this work differed between viruses (Table 5), ranging from 14.5% for YaV1 to 100% for macluraviruses. Low sanitation success rates were obtained for badnaviruses (30.1%), YMMV (24.9%) and potexviruses (18.9%). 

## 4. Discussion

Sexual reproduction provides natural sanitation for most plant viruses, which are not seed transmitted [49]. However, vegetatively propagated crops are deprived of this natural sanitation and accumulate viruses over time, hampering control strategies based on the production and use of clean seeds [50]. This paper reports on the development and optimization of diagnostic tools for the detection of viruses infecting yams and their successful implementation for the sanitation of yam accessions, resulting in virus-free yam germplasm and paving the way to a better control of yam-infecting viruses.

A comprehensive diagnostic strategy was designed in order to increase sensitivity and to reduce costs and labor. Extraction of high quality TNAs from yam was achieved using the method developed by Foissac et al. [35]. This method is based on the use of silica in order to remove polyphenolic and mucilaginous compounds, which are naturally present in high amounts in yam and hamper the purification of high quality nucleic acids and their long-term conservation [51]. Universal primers were used to prime the synthesis of cDNAs from TNAs, resulting in versatile templates that could be used for the detection of all yam viruses by PCR, except for badnaviruses, which were detected by M-IC-PCR to avoid false positives resulting from the presence of eDBVs in the genome of yams. Existing primers were used for the detection of CMV, YaV1, YMV, YMMV and badnaviruses, whereas new primers were designed for the detection of more recently characterized viruses such as DMaV, macluraviruses and potexviruses (Table 3). PCR conditions were optimized for all primer combinations used in this work, resulting in accurate sensitivity and specificity. HTS performed on sanitized plants confirmed the results obtained by RT-PCR-based tests, demonstrating the accuracy of the molecular detection tools developed in this study. However, although the use of dsRNA templates for HTS proved successful for the detection of RNA viruses in yam [13], it is not appropriate for the detection of DNA viruses. Therefore, the presence in the analyzed sanitized plants of residual badnaviruses below the threshold of detection by M-IC-PCR cannot be ruled out.

Several potexviruses have been reported in yams [15]. Degenerate primers designed by van der Vlugt and Berendsen [31] were successfully used to detect potexviruses in several crops. However, these primers generated non-specific amplification products when used in yams, resulting in unreliable diagnosis. Based on the alignment of nucleotide sequences in the polymerase domain of all known yam potexviruses, a new set of primers was designed (Table 3) and used in RT-nested-PCR experiments, providing accurate, reliable and reproducible diagnosis. Such nested PCR is successfully used for the detection of other viruses in the families *Alphaflexiviridae* and *Betaflexiviridae,* whose members are difficult to detect by RT-PCR [35,52]. Indexing confirmed that only one of the three potexviruses reported in yams, yam virus X, is present in the BRC-TP yam collection [15].

Taking advantage of the above-mentioned comprehensive diagnostic tools, we undertook the first large-scale virus prevalence study in a yam germplasm collection. The indexing of 396 accessions conserved by the BRC-TP showed that 91.6% (363/396) of these accessions were infected by at least one virus and 71.5% of them were co-infected by two to five viruses, demonstrating that viruses could easily coexist in the yam plant (Figure 3B and Appendix A).

Potyviruses displayed the highest prevalence in the analyzed yam collection, with 71% of the accessions (281/396) infected by YMV and/or YMMV (Figure 3A and Appendix A). Prevalence of potyviruses was 60% in *D. alata*, 25% in *D. bulbifera*, 73.5% in *D. cayenensis-rotundata*, 87.5% in *D. esculenta* and 79.1% in *D. trifida*. Potyviruses cause severe damage in a wide range of crops [53] including yams [54], therefore such a high infection rate threatens the long term conservation of the BRC-TP yam collection. 

Surprisingly, none of the accessions were infected by CMV, although this virus has one of the broadest host spectrum ranges among plant viruses [55]. CMV was detected in yam in sub-Saharan Africa with a low prevalence [56,57], and viral particles of CMV were reported in yam in Guadeloupe [16]. However, the virus could not be detected by molecular tests despite numerous attempts using several sets of primers, suggesting that isometric particles of another yam-infecting virus such as DMaV were confused with CMV virions or that available molecular detection tools are not able to detect CMV isolates that may be present in Guadeloupe.

Significant differences in virus prevalence were observed between yam species (Figure 3A and Table 5). For example, *D. alata* accessions were mostly infected by YaV1 and/or YMMV, whereas *D. cayenensis-rotundata* accessions were mostly infected by YMV, *D. esculenta* by macluraviruses and/or YaV1, and *D. trifida* by YMMV, badnaviruses and/or YMV. Overall, YMV was mostly detected in *D. cayenensis-rotundata* and *D. trifida* accessions, YMMV in *D. alata* and *D. trifida* accessions, YaV1 in *D. alata* and *D. esculenta* accessions, potexviruses in *D. trifida* accessions and macluravirus in *D. esculenta* accessions (Table 5). On the contrary, macluraviruses were not detected in *D. cayenensis-rotundata* and *D. trifida* accessions, and prevalence of DMaV and badnaviruses was similar among yam species.

The sanitation method described in this work involves a combination of thermotherapy and meristem culture. Thermotherapy allows the reduction of the viral replication rate in vascularized tissues and meristem excision favors the regeneration of healthy plantlets from virus-free totipotent cells. This combination proved moderately efficient on the vitroplants used in this work, since only 36 out of 351 generated meriplants (9.7%) were fully sanitized, representing 16 of the 57 accessions subjected to sanitation (Table 4). Comparatively, 100% and 52.4% of sweet potato plantlets infected by sweet potato feathery mottle virus (SPFMV, genus *Potyvirus*) and sweet potato leaf curl virus (SPLCV, genus *Begomovirus*), respectively, were sanitized by the combination of thermotherapy and meristem excision [58,59]. The low rate of sanitation success in yam may result from the small size of yam meristems, which are particularly difficult to excise without surrounding contaminated cells, even following thermotherapy (Figure 2B). Our work also shows that the efficiency of the sanitation process varies between viruses, ranging from 14.5% for YaV1 to 100% for macluraviruses (Table 5). This difference could result from differential sensitivities to high temperatures among yam-infecting viruses and/or their cellular localization.

It is important to stress out that the meriplants considered as virus-free in our work are sanitized only for the viruses for which optimized diagnosis tools are currently available, with the exception of the four plants assessed by HTS. Therefore, the sanitation scheme reported in this paper does not yet include recently characterized yam viruses such as AiPoV1, DMV and YVY, for which molecular diagnostic tools are being developed. However, the sanitation process reported in this work (Figure 1) has been designed to accommodate the diagnosis of additional viruses upon the implementation of detection tools targeting these viruses. This process is therefore similar in essence to more global initiatives such as the one developed for other vegetatively propagated crops such as *Rubus* spp [60]. However, the approach we developed purposely does not include the systematic use of HTS in our evaluation pipeline (Figure 1) because the methods and approaches described in this paper are meant to be used in yam germplasm repositories that are often located in developing countries where access to HTS is difficult and expensive. Overall, the use of HTS for routine diagnosis on a large number of plants is not cost effective and should be reserved for confirming the viral status at the very end of the sanitation process, as suggested by Bömer et al. [61].

Sanitation is a long and time-consuming process: a complete sanitation cycle takes up to nine months, not including the whole plant verification step (Figure 1). Hence, sanitation should be streamlined in order to focus on pathogenic viruses. Meanwhile, an increasing number of novel viruses are reported in yam, based on HTS data [5,13,62]. There is a need to assess their symptomatology, biology, epidemiology and impact on yam production in order to decide whether they are worth including in sanitation programs. Replacing thermotherapy by cryotherapy could also help alleviating sanitation costs and labor since cryotherapy consists in instant deep freezing of bud tissues and results in the destruction of infected vascularized tissues surrounding virus-free meristematic cells, thus enabling the regeneration of sanitized plantlets [63]. When applied to axillary buds of white yam (*Dioscorea rotundata* Poir.), this approach led to the regeneration of 70% virus-free plants from YMV-infected plants [14]. This sanitation success rate is very similar to that obtained by the method described in this paper (62.7%) therefore cryotherapy should be evaluated on other yam species and yam-infecting viruses.

## Figures and Tables

**Figure 1 viruses-12-01101-f001:**
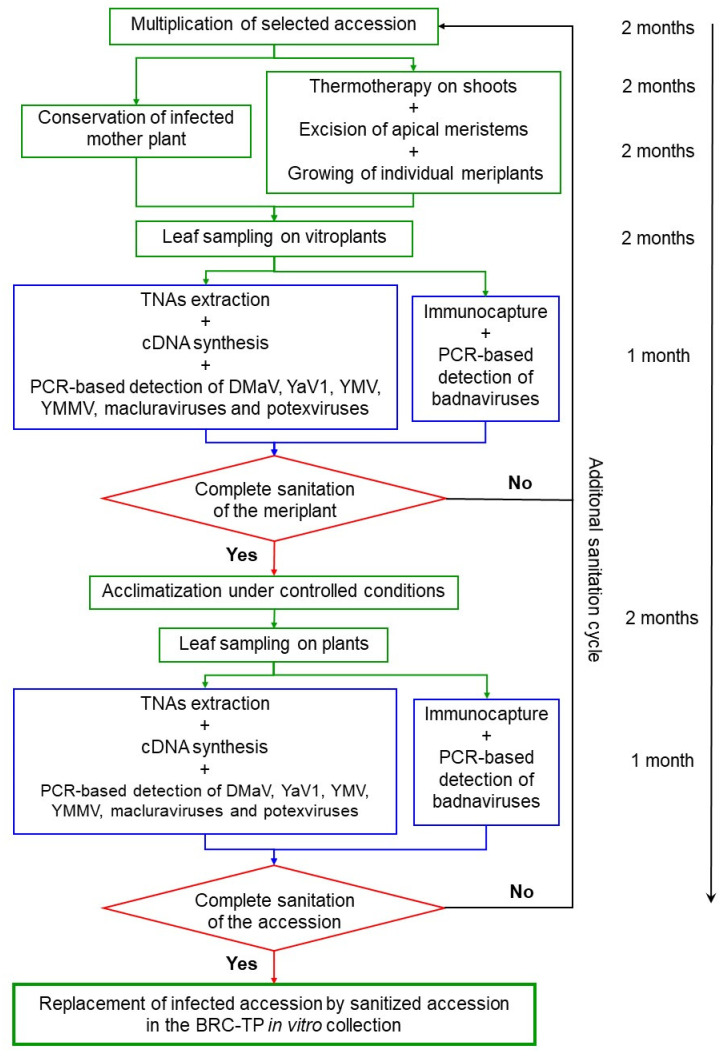
Flowchart of the sanitation process used in this work. Green boxes represent steps carried out *in vitro* or in controlled conditions; blue boxes represent molecular diagnosis steps; red diamonds represent decision steps. A timeline is provided. TNAs: total nucleic acids.

**Figure 2 viruses-12-01101-f002:**
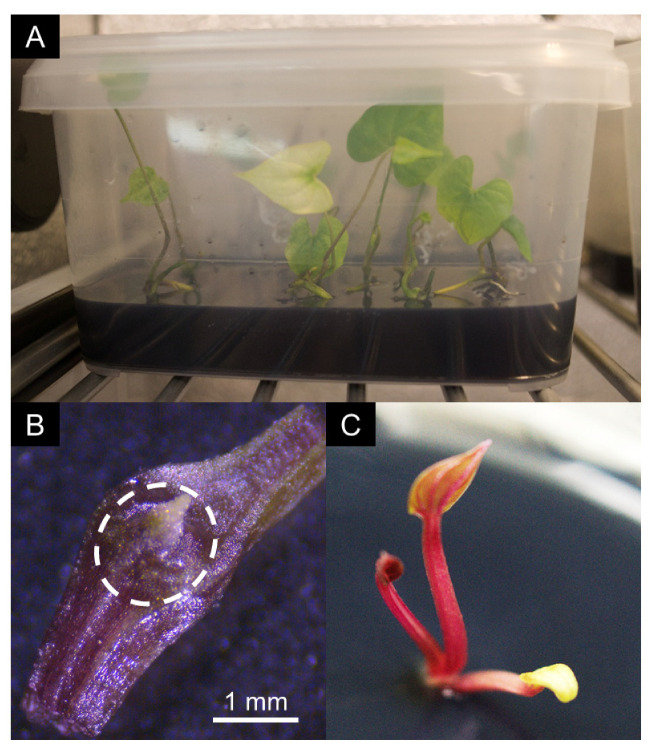
Illustration of *in vitro* steps of the sanitation process. (**A**) *Dioscorea trifida* cuttings inside a thermal chamber at 34 °C. (**B**) Explant of *D. trifida*. The dotted white circle highlights the meristematic bud with leaf primordia. Scale of size is provided. (**C**) Plantlet of *D. alata* obtained from meristematic culture, called meriplant.

**Figure 3 viruses-12-01101-f003:**
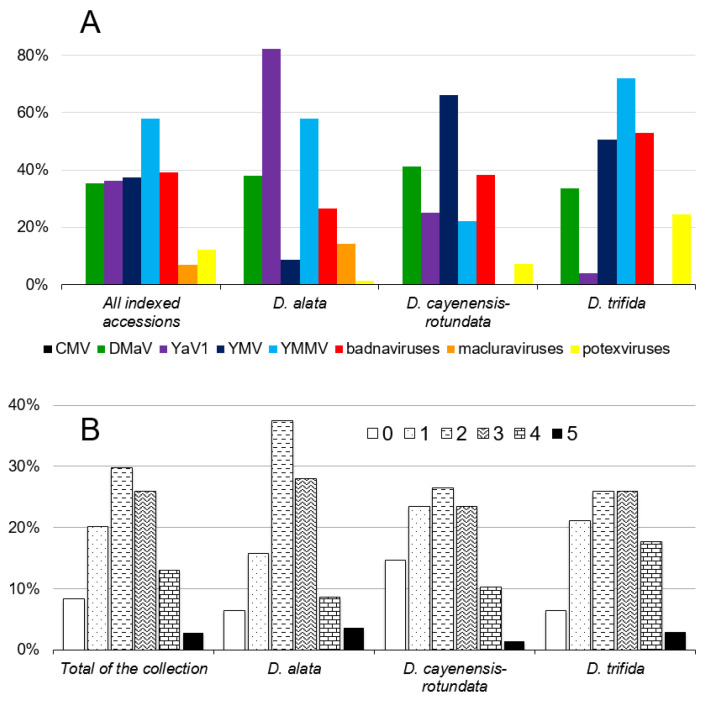
Results of viral indexing of the 396 tested accessions and the three most represented yam species (*D. alata*, *D. cayenensis-rotundata* and *D. trifida*) from the BRC-TP *in vitro* yam collection. (**A**) Prevalence of the eight yam viruses reported in Guadeloupe. CMV: cucumber mosaic virus; DMaV: Dioscorea mosaic associated virus; YaV1: yam asymptomatic virus 1; YMV: yam mosaic virus; YMMV: yam mild mosaic virus. (**B**) Percentage of accessions infected by 0, 1, 2, 3, 4 or 5 distinct viruses.

**Table 1 viruses-12-01101-t001:** List of yam-infecting viral species identified in Guadeloupe.

Family	Genus	Species	Genbank Accession Number of Complete Genome	Reference for Presence of the Virus in Guadeloupe
*Alphaflexiviridae*	*Potexvirus*	*Yam virus X* (YVX)	KJ711908	[15]
		yam potexvirus 1	Not determined	
*Bromoviridae*	*Cucumovirus*	*Cucumber mosaic virus* (CMV)	Not determined	[16]
*Caulimoviridae*	*Badnavirus*	*Dioscorea bacilliform AL virus* (DBALV)	KX008573	[9]
		*Dioscorea bacilliform TR virus* (DBTRV)	KX430257	
*Closteroviridae*	*Ampelovirus*	yam asymptomatic virus 1 (YaV1)	MT409627	[13]
*Potyviridae*	*Macluravirus*	Dioscorea alata macluravirus	Not determined	[17]
		Dioscorea esculenta macluravirus	Not determined	
*Potyviridae*	*Potyvirus*	*Yam mild mosaic virus* (YMMV)	JX470965	[18]
		*Yam mosaic virus* (YMV)	U42596	[19]
*Secoviridae*	*Sadwavirus*	*Dioscorea mosaic associated virus* (DMaV)	KU215538, KU215539	[20]

**Table 2 viruses-12-01101-t002:** Composition of the two media used in the sanitation process.

	Growth Medium		Regeneration Medium
Components	*D. alata* and *D. cayenensis-rotundata*	*D. trifida*	
MacroMS	^a^	^a^	^b^
MicroMS	^a^	^a^	^a^
FeEDTA	^a^	^a^	^a^
Morel vitamins ^b^	^c^	^c^	^c^
Sucrose	30 g/L	30 g/L	30 g/L
L-Glutamine	200 mg/L	200 mg/L	-
L-Glycine	-	100 mg/L	-
Cysteine	50 mg/L	-	-
BAP	1 mg/L	100 µg/L	100 µg/L
Agar	7 g/L	7 g/L	7 g
Activated charcoal	2 g/L	2 g/L	2 g

^a^ Composition described by Murashige and Skoog [26]; ^b^ Composition described by Murashige and Skoog [26], with quantity of components divided by two; ^c^ Composition described by Morel [27].

**Table 3 viruses-12-01101-t003:** Sequence of primers and PCR parameters used in this study.

Name	Targeted Virus or Gene	Sequence	Reference	[Primers] (nM)	[MgCl_2_] (mM)	Annealing Temperature (°C)	Size of Amplification Product
CMV1-F	CMV ^a^	5′-GTA GAC ATC TGT GAC GCG A-3′	[28]	400	1.5	55	540 bp
CMV1-R		5′-GCG CGA AAC AAG CTT CTT ATC-3′					
Seco-1F	DMaV ^a^	5′ AAC TCC WTC WGG WTT YGC TYT GAC 3′	this study	400	2	46	323 bp
Seco-1R		5′ CCC ACT TYC TYT TGA GAA AAT CAA 3′					
DiosClos-F	YaV1 ^a^	5′ CTC TTT AGG TTT CCC ATT TAT CA 3′	[13]	800	2	49	285 bp
DiosClos-R		5′ TGG TTC TAC ATT ACT AGA CTA C 3′					
YMV1	YMV ^a^	5′ TGC GGA ACT CRA AAG AAC 3′	[29]	200	3	53	196 bp
YMV2		5′ TGC CAT CAA ATC CAA ACA 3′					
YMMV CP 2F	YMMV ^a^	5′ GGC ACA CAT GCA AAT GAA AGC 3′	[30]	300	3	53	249 bp
YMMV UTR 1R		5′ CAC CAG TAG AGT GAA CAT AG 3′					
YamMac4F	Yam macluraviruses ^b^	5′ CHG CAG CWA TYG GKM GTG 3′	this study	600	2	47	292 bp
YamMac5R		5′ GGG TTG CTG AGC RTY GGA 3′					
Potex-2RC	Potexviruses ^c^	5′ AGC ATR GCN SCR TCY TG 3′	[31]	400	2	49	584 bp
Potex-5		5′ CAY CAR CAR GCM AAR GAY GA 3′					
YamX-1F	Yam	5′ GAY ATT GGR GAY RTW YTI TTY WKG 3′					
YamX-2F	Potexviruses ^b^	5′ GRC ART CYC CAG ATT TYR AYH YRM A 3′					
YamX-3F ^d^		5′ CIC ART GGG TIA AGA AAR WKG A 3′	this study	300	2	43	255 bp
YamX-4R		5′ CAT IAC AGT TTS TTG CAT RAA IG 3′					
YamX-5R ^d^		5′ GRT CRA AIG CIG TRA ART CAT TIG C 3′					
BadnaFP	Badnaviruses ^c^	5′ ATG CCI TTY GGI ITI AAR AAY GCI CC 3′	[32]	200	3	49	579 bp
BadnaRP		5′ CCA YTT RCA IAC ISC ICC CCA ICC 3′					
atpB1	*atpB* plastid gene	5′ GGT CCT ATG ATT TCC ACA C 3′	[33]	200	3	49	1462 pb
atpB2		5′ CTT CAA TTT GTT CTC CAC 3′					

^a^ Specific detection primers; ^b^ Generic detection primers for yam viruses; ^c^ Generic detection primers; ^d^ Selected primers for detection of yam potexviruses.

**Table 4 viruses-12-01101-t004:** List of sanitized meriplants.

Common Name	Voucher ID	Yam Species	Meriplants ^a^	Eliminated Viruses ^b^
Pacala station	PT-IG-00017	*D. alata*	35/TM1/TM1 ^c^	DMaV + YaV1 + macluraviruses
Oriental	PT-IG-00074	*D. alata*	93/TM1	YaV1 + macluraviruses
			93/TM5	
Toro	PT-IG-00078	*D. alata*	97/TM2	YaV1 + YMMV + macluraviruses
MH 22	PT-IG-00164	*D. trifida*	278/TM1 ^d^	DMaV + YMMV
2.22	PT-IG-00244	*D. trifida*	281/TM1 ^d^	YMMV
			281/TM2	
Amoumbé 1	PT-IG-00425	*D. trifida*	489/TM3/TM2 ^c^	YMV + YMMV + badnaviruses
			489/TM3/TM3 ^c^	
G. O 18	PT-IG-00450	*D. trifida*	514/TM7	YMMV
			514/TM8	
			514/TM10	
			514/TM11	
			514/TM12	
Jossaud 2	PT-IG-00453	*D. trifida*	517/TM1	DMaV + YMMV + badnaviruses
Marché Cayenne	PT-IG-00481	*D. trifida*	528/TM2 ^d^	YMV
109 RA	PT-IG-00531	*D. cayenensis-rotundata*	595/TM1	YMV + badnaviruses
			595/TM2	
			595/TM3	
			595/TM4	
			595/TM5	
			595/TM6	
			595/TM7	
			595/TM8	
Tahiti couleuvre	PT-IG-00557	*D. alata*	621/TM3	YaV1 + YMMV
3.45	PT-IG-00585	*D. trifida*	650/TM1 ^d^	YMV + YMMV + badnaviruses
SM 23	PT-IG-00650	*D. trifida*	715/TM2	DMaV + YMV + YMMV
Apatou 3	PT-IG-00653	*D. trifida*	718/TM1	YMMV
			718/TM2	
Saül 36	PT-IG-00662	*D. trifida*	727/TM5	badnaviruses
St Jean 9	PT-IG-00681	*D. trifida*	746/TM2	YMMV
			746/TM4	
			746/TM5	
			746/TM7	
			746/TM8	
			746/TM9	

^a^ The code of meriplant name is composed by the number of the accession followed by “T”, meaning thermotherapy, “M”, meaning meristem culture, and the number of the generated meriplant; ^b^ According to the indexation of the mother plant (Appendix A); ^c^ These meriplants were sanitized after two cycles of sanitation; ^d^ the sanitation of these meriplants was confirmed by high throughput sequencing.

**Table 5 viruses-12-01101-t005:** Prevalence and sanitation rate of targeted viruses.

Yam Species	Badnaviruses	CMV	DMaV	Macluraviruses	Potexviruses	YaV1	YMV	YMMV
*D. alata*	26.6% (37/139) ^a^	0.0% (0/140)	37.9% (53/140)	14.3% (20/140)	1.4% (2/140)	82.1% (115/140)	8.6% (12/140)	57.9% (81/140)
*D. bulbifera*	25.0% (1/4)	0.0% (0/4)	25.0% (1/4)	25.0% (1/4)	0.0% (0/4)	0.0% (0/4)	25.0% (1/4)	0.0% (0/4)
*D. cayenensis-rotundata*	38.2% (26/68)	0.0% (0/68)	41.2% (28/68)	0.0% (0/68)	7.4% (5/68)	25.0% (17/68)	66,2% (45/68)	22,1% (15/68)
*D. esculenta*	0.0% (0/8)	0.0% (0/8)	0.0% (0/8)	87.5% (7/8)	0.0% (0/8)	62.5% (5/8)	25.0% (2/8)	87.5% (7/8)
*D. trifida*	52.9% (90/170)	0.0% (0/172)	33.7% (58/172)	0.0% (0/172)	24.4% (42/172)	4. 1% (7/172)	50.6% (87/140)	72.1% (124/172)
Sanitation success rate	30.1% (40/133) ^b^	Na ^c^	40.0% (50/122)	100.0% (37/37)	18.9% (17/90)	14.5% (10/69)	62.4% (53/85)	24.9% (75/301)

^a^ Percentage of infected accessions (number of infected accessions/total number of indexed accessions); ^b^ Percentage of sanitized meriplants (number of sanitized meriplants/total number of meriplants generated from infected accessions); ^c^ Not applicable.

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
