# Peer review of "Molecular Viral Diagnosis and Sanitation of Yam Genetic Resources: Implications for Safe Yam Germplasm Exchange"

_viruses, 2020, doi:10.3390/v12101101_

Round 1
Reviewer 1 Report
This work describes i) the development of primers for the detection of select viruses infecting yam in Guadeloupe ii) a thermal therapy/meristem excision approach for virus elimination, and iii) an evaluation of this system on a yam germplasm repository in Guadeloupe.
The work introduces the reader to a long list of new viruses infecting yam, but surprisingly, does not address any of these viruses in their sanitation efforts to generate clean germplasm. Out of an abundance of caution, we routinely test our foundation germplasm for viruses and fastidious bacteria that are not known to occur in our geographic locale. With new viruses becoming discovered in yam, how can the authors be sure their germplasm is free of these pathogens? Have all of the accessions been tested for these 13 "new" yam viruses?
There are "clean" plant systems being developed all over the world, including how to evaluate this germplasm using the powerful sequencing technologies that are now widely available. Yet there is little mentioned on this topic and how this yam program compares and contrasts with these existing systems. I think it would be helpful for the authors to explain why their approach (Figure 2) is being adopted, and how it compares to those previously proposed (such as Gergerich et al 2015). More discussion on how HTS can impact this clean yam germplasm is needed. I did not see mention of the work by Bomer et al 2019 that addressed the detection of yam viruses by HTS (NGS in their publication).
The authors do use HTS to evaluate and confirm their PCR-based results, but it is unclear if HTS will be used in their evaluation pipeline. HTS is absent from Figure 2. The second surprise for me was that the HTS data indicated a lack of viral reads. My first issue with this statement is that no metrics are provided for the HTS. How many reads were generated for each multiplexed sample? How do we know there was not a technical issue with the dsRNA extraction, resulting in a false negative. The authors need to elaborate on their methodology and explain why their template molecule (dsRNA) was used instead of sRNA or enriched total RNA (I am not suggesting dsRNA is a inappropriate template, but rather justification for its use). In my experience, there are almost always viral reads in an HTS dataset, even from "clean" plants. These are often myco-like viruses or some form of contaminant. In this work, "spiking" a biological sample with a known positive sample to ensure the "wet lab" techniques are working appropriately, as well as spiking an HTS data set with viral sequences to ensure in silico discovery techniques are working, would be highly appropriate and instill confidence in the findings.
The manuscript is generally well written and easy to understand. There is a lack of italicizing plant species names, viral genera, and latin phrases that needs to be addressed throughout the work. Overall I believe the work is important and can be published if the above issues are addressed. A resource for yam germplasm that is free of targeted viruses is of great importance to agriculture not only in Guadeloupe, but other yam-growing nations.
In section 3.1, no actual results are provided for the badnavirus work; only an explanation of why the work was done. What are the controls for this and other PCR-based assays?
Other comments of a minor nature are listed below:
L33: perhaps replace "first" with "primary"
L37: perhaps replace "more" with "longer"
L40: How can this species be endemic if it is from the Amazon?
L78: perhaps "...sanitation program needs to be assessed." Some reports find that ELISA is satisfactory for the detection of certain viruses in clean germplasm.
Figure 1 is probably not critical and could be moved to the supplementary data
L58 (page 6 it appears the line numbers reset): replace "displayed" with perhaps either "contained" or "harboured"
Perhaps the use of "wean" is appropriate, but I would think this verb could be replaced by "acclimatized" or something similar. I have not heard this word used in plant tissue culture before.
Figure 2: use "cDNA" in flowchart
Figure 2: can an approximate timeline be provided for this figure; in the discussion it is mentioned this takes about 9 months. It would be helpful to see how much time is taken at each step.
Figure 3: I did not see a scale in B)
L113 (page 9): Replace "On the opposite" with "In contrast" here and elsewhere.
L163: "...accessions were infected by two..."
L183: replace "average" with "intermediate"
L185: I would still recommend testing for CMV and other targeted viruses, regardless of their absence in a location
L200: 20 leaf stage
L200: does this mean both of the pair of cloned meriplants died? That seems unusual.
L25 (discussion): "non-specific"
L30: replace or eliminate "straight"
L57-60: this seems to be a rewording of the previous sentences in this paragraph
Author Response
Answers to comments of Reviewer #1
The work introduces the reader to a long list of new viruses infecting yam, but surprisingly, does not address any of these viruses in their sanitation efforts to generate clean germplasm. Out of an abundance of caution, we routinely test our foundation germplasm for viruses and fastidious bacteria that are not known to occur in our geographic locale.
Authors’ answer: Our article presents the results accumulated since the beginning of the sanitation process implemented at the Tropical Plant Biological Resource Center in 2014. At that time, the 13 new viruses infecting yam that are presented in the introduction had not yet been discovered. Therefore we initially based our diagnostic scheme on the viruses known to occur in yam (CMV, YMV, YMMV, potexviruses, badnaviruses) and on those we had identified ourselves (YaV1, DMaV, macluraviruses). We expanded the range of viruses tested to include newly described yam viruses upon their publication in the literature. Nowadays, just like Reviewer #1 does for his collections, we systematically test our germplasm for all the viruses listed in the introduction, including AiPoV1, YVY and potyviruses. The text in the introduction and discussion sections (page XX, lanes XX) has been modified Sentences have been added in the introduction (l. 70 and 74) and discussion (l. 80-85) to explain this.
With new viruses becoming discovered in yam, how can the authors be sure their germplasm is free of these pathogens?
Authors’ answer : We agree with reviewer #1 and are fully aware of the current limitations of our diagnosis scheme in terms of the number of viruses tested. The text in the discussion section has been modified to accommodate the comment of reviewer #1 and a paragraph has been added (page XX, lane XX).
Have all of the accessions been tested for these 13 "new" yam viruses?
Authors’ answer : All of the accessions have not been tested for the 13 new yam viruses yet because diagnostic tools are not available for all of them. Hence we are developing the tools that are still missing, using the strategy outlined in our paper. The sanitized meriplants reported in our paper will be assessed using these new detection tools. The text in the discussion section has been modified to address this comment.
There are "clean" plant systems being developed all over the world, including how to evaluate this germplasm using the powerful sequencing technologies that are now widely available. Yet there is little mentioned on this topic and how this yam program compares and contrasts with these existing systems. I think it would be helpful for the authors to explain why their approach (Figure 2) is being adopted, and how it compares to those previously proposed (such as Gergerich et al 2015).
Authors’ answer : Our work is still at a very early stage of a global scheme for the production of healthy seedlings in order to provide clean material to farmers. Our scheme (Figure 2) is thus a small part of the global scheme presented by Gergerich et al (2015). Thus, the next step of our work will be to develop a system of production and distribution of our sanitized yam seedlings to producers, with monitoring protocols similar to those cited by Gergerich. Moreover, the conservation of our collection in the form of vitroplants allows us to avoid a regular monitoring of the contamination of our sanitized plants, but imposes an additional verification step on whole plants. A sentence is added in the discussion to explain this difference (l. XX).
More discussion on how HTS can impact this clean yam germplasm is needed.
Authors’ answer We feel that HTS is not appropriate for routine diagnosis on large numbers of plants and should rather be reserved for confirming the viral status of sanitized elite germplasm at the very end of the sanitation process. The text of the discussion section has been modified to address this comment; a paragraph has been added to discuss the use of HTS.
I did not see mention of the work by Bomer et al 2019 that addressed the detection of yam viruses by HTS (NGS in their publication).
Authors’ answer : The text has been modified to address this comment, and the paper of Bömer et al., 2019 is now cited in the discussion section (l. 89).
The authors do use HTS to evaluate and confirm their PCR-based results, but it is unclear if HTS will be used in their evaluation pipeline. HTS is absent from Figure 2.
Authors’ answer HTS is not used currently in our evaluation pipeline, but it should be. We hope to be able to implement it soon.
The second surprise for me was that the HTS data indicated a lack of viral reads. My first issue with this statement is that no metrics are provided for the HTS. How many reads were generated for each multiplexed sample? How do we know there was not a technical issue with the dsRNA extraction, resulting in a false negative. The authors need to elaborate on their methodology and explain why their template molecule (dsRNA) was used instead of sRNA or enriched total RNA (I am not suggesting dsRNA is a inappropriate template, but rather justification for its use). In my experience, there are almost always viral reads in an HTS dataset, even from "clean" plants. These are often myco-like viruses or some form of contaminant. In this work, "spiking" a biological sample with a known positive sample to ensure the "wet lab" techniques are working appropriately, as well as spiking an HTS data set with viral sequences to ensure in silico discovery techniques are working, would be highly appropriate and instill confidence in the findings.
Authors’ answer
The manuscript is generally well written and easy to understand. There is a lack of italicizing plant species names, viral genera, and latin phrases that needs to be addressed throughout the work.
Authors’ answer : Italics were removed by the editing process. They have now been restored.
Overall I believe the work is important and can be published if the above issues are addressed. A resource for yam germplasm that is free of targeted viruses is of great importance to agriculture not only in Guadeloupe, but other yam-growing nations.
Authors’ answer : We agree with this statement and we thank reviewer #1 for acknowledging the importance of our work.
In section 3.1, no actual results are provided for the badnavirus work; only an explanation of why the work was done. What are the controls for this and other PCR-based assays?
Authors’ answer : M-IC-PCR assay is used for the routine diagnosis of yam badnaviruses, as published previously (Umber et al., 2017). All detection tests were carried out with appropriate controls. The text in the material and method section has been modified to address tis comment (l. 24-25 and l. 40-41).
Other comments of a minor nature are listed below:
L33: perhaps replace "first" with "primary"
Authors’ answer : “first” was replaced by “largest”.
L37: perhaps replace "more" with "longer"
Authors’ answer : Change made.
L40: How can this species be endemic if it is from the Amazon?
Authors’ answer : Indeed, this species is endemic to the American continent, as opposed to other species that originate from Asia or Africa. Therefore “endemic” was replaced by “American”.
L78: perhaps "...sanitation program needs to be assessed." Some reports find that ELISA is satisfactory for the detection of certain viruses in clean germplasm.
Authors’ answer : The text was modified to address this comment.
Figure 1 is probably not critical and could be moved to the supplementary data
Authors’ answer : Figure was moved to the supplementary data.
L58 (page 6 it appears the line numbers reset): replace "displayed" with perhaps either "contained" or "harboured"
Authors’ answer : “displayed” was replaced by “harboured”
Perhaps the use of "wean" is appropriate, but I would think this verb could be replaced by "acclimatized" or something similar. I have not heard this word used in plant tissue culture before.
“weaned” was replaces by “acclimatized” throughout the text.
Figure 2: use "cDNA" in flowchart
Authors’ answer : Change made.
Figure 2: can an approximate timeline be provided for this figure; in the discussion it is mentioned this takes about 9 months. It would be helpful to see how much time is taken at each step.
Authors’ answer : A timeline was added in the figure.
Figure 3: I did not see a scale in B
Authors’ answer : The scale was erased during the editing process. It has now be restored.
L113 (page 9): Replace "On the opposite" with "In contrast" here and elsewhere.
Authors’ answer : Changes made throughout the text.
L163: "...accessions were infected by two..."
Authors: Change made.
L183: replace "average" with "intermediate"
Authors’ answer : Change made.
L185: I would still recommend testing for CMV and other targeted viruses, regardless of their absence in a location
Authors’ answer : This recommendation has been taken into account. We now test for CMV in all prevalence or diversity studies.
L200: 20 leaf stage
Authors’ answer : Change made.
L200: does this mean both of the pair of cloned meriplants died? That seems unusual.
Authors’ answer : In fact, the vitroplants died before they could be acclimatized, due to contaminations by fungi during cell culture.
L25 (discussion): "non-specific"
Authors’ answer : Change made.
L30: replace or eliminate "straight"
Authors’ answer : Change made.
L57-60: this seems to be a rewording of the previous sentences in this paragraph
Authors’ answer : The redundant sentence was removed from the text.
Reviewer 2 Report
This was an interesting paper which I consider to be worthy of publication. My comments regarding the improvement of the manuscript are noted below:
line 19 and 66: Centre v Center
line 28: replace diffusion with dissemination
line 33: Nigeria being the highest producer
line 60: replace “utterly necessary” with “critical”
line 78: all eight viruses occurring in yams in Guadeloupe - but Table 1 lists 11 viruses. Please clarify.
line 89: D. nummularia
line 91: for consistency, the headings “growth medium” and “regeneration medium” in Table 2 should start with capitals; L-Glutamine; Cysteine; should define BAP; in regeneration medium, the amounts of Agar and Activated charcoal should both be 7 g/L
line 94: ….yam macluravirus viral genomic
page 5, line 6: products raised generated from….
Line 32: one hour replaced with 1 h
Lines 37, 46 and 143: MgCl2
Line 84: No scale size is provided in Figure 3
Line 86: dotted
Line 88: There are two full stops at the end of the sentence
Line 91: please define dish soap or replace with more appropriate terminology
Line 113 and 157: replace “on the opposite” with “in contrast”
Line 154: Since CMV is a very common virus, was it not possible to obtain a known infected sample to use as a positive control?
Line 167: I can’t see the point of including CMV in Fig. 5 considering it was not detected.
Page 15, line 15: replace “raised” with “generated”
Line 45: broadest host spectrum range
Line 166: Triennial not triennal
Did the authors consider the use of RCA to theoretically detect a broader range of viruses with circular DNA genomes?
Author Response
line 19 and 66: Centre v Center
Authors’ answer : Change made.
line 28: replace diffusion with dissemination
Authors’ answer : Change made.
line 33: Nigeria being the highest producer
Authors’ answer: “largest” preferred to “highest”.
line 60: replace “utterly necessary” with “critical”
Authors’ answer: Change made.
line 78: all eight viruses occurring in yams in Guadeloupe - but Table 1 lists 11 viruses. Please clarify.
Authors’ answer: Table 1 presents the 11 virus species that have been identified in Guadeloupe. Some of them are detected simultaneously using generic tests (badnaviruses, potexviruses and macluraviruses). The text has been modified to avoid confusion.
line 89: D. nummularia
Authors’ answer: Change made.
line 91: for consistency, the headings “growth medium” and “regeneration medium” in Table 2 should start with capitals; L-Glutamine; Cysteine; should define BAP; in regeneration medium, the amounts of Agar and Activated charcoal should both be 7 g/L
Authors’ answer: Changes made, except for the amounts of agar and activated charcoal that are accurate.
line 94: ….yam macluravirus viral genomic
Authors’ answer: Change made
page 5, line 6: products raised generated from….
Authors’ answer: Change made
Line 32: one hour replaced with 1 h
Authors’ answer: Change made
Lines 37, 46 and 143: MgCl2
Authors’ answer: Change made
Line 84: No scale size is provided in Figure 3
Authors: The scale was erased during the editing process. It has been restored.
Line 86: dotted
Authors’ answer: Change made
Line 88: There are two full stops at the end of the sentence
Authors’ answer: Change made
Line 91: please define dish soap or replace with more appropriate terminology
Authors’ answer: “dish soap” was replaced by “detergent”
Line 113 and 157: replace “on the opposite” with “in contrast”
Authors’ answer: Change made
Line 154: Since CMV is a very common virus, was it not possible to obtain a known infected sample to use as a positive control?
Authors’ answer: We used a positive control but could not detect CMV in any of our yam samples.
Line 167: I can’t see the point of including CMV in Fig. 5 considering it was not detected.
Authors’ answer: CMV was purposely included in Fig. 5 to show that it could not be detected in any of the samples.
Page 15, line 15: replace “raised” with “generated”
Authors’ answer: Change made
Line 45: broadest host spectrum range
Authors’ answer: Change made
Line 166: Triennial not triennial
Authors’ answer: Changes made
Did the authors consider the use of RCA to theoretically detect a broader range of viruses with circular DNA genomes?
Authors’ answer: Yes. RCA has the potential to detect novel yam badnaviruses, as exemplified by the work published by Bömer et al in 2016. We plan to assess RCA in our laboratory, and depending on its efficiency, include its use in our sanitation scheme.
Round 2
Reviewer 1 Report
I appreciate the authors response to my comments. However there does not seem to be a response or revision to the manuscript regarding one of my major concerns:
The second surprise for me was that the HTS data indicated a lack of viral reads. My first issue with this statement is that no metrics are provided for the HTS. How many reads were generated for each multiplexed sample? How do we know there was not a technical issue with the dsRNA extraction, resulting in a false negative. The authors need to elaborate on their methodology and explain why their template molecule (dsRNA) was used instead of sRNA or enriched total RNA (I am not suggesting dsRNA is a inappropriate template, but rather justification for its use). In my experience, there are almost always viral reads in an HTS dataset, even from "clean" plants. These are often myco-like viruses or some form of contaminant. In this work, "spiking" a biological sample with a known positive sample to ensure the "wet lab" techniques are working appropriately, as well as spiking an HTS data set with viral sequences to ensure in silico discovery techniques are working, would be highly appropriate and instill confidence in the findings.
